# Ubiquitin-Specific Proteases as Potential Therapeutic Targets in Bladder Cancer—In Vitro Evaluation of Degrasyn and PR-619 Activity Using Human and Canine Models

**DOI:** 10.3390/biomedicines11030759

**Published:** 2023-03-02

**Authors:** Łukasz Nowak, Wojciech Krajewski, Ewa Dejnaka, Bartosz Małkiewicz, Tomasz Szydełko, Aleksandra Pawlak

**Affiliations:** 1Department of Minimally Invasive and Robotic Urology, University Center of Excellence in Urology, Wroclaw Medical University, 50-556 Wroclaw, Poland; 2Department of Pharmacology and Toxicology, Faculty of Veterinary Medicine, Wroclaw University of Environmental and Life Sciences, 50-375 Wroclaw, Poland

**Keywords:** bladder cancer, urothelial carcinoma, ubiquitin-specific proteases, Degrasyn, PR-619

## Abstract

Background: The inhibition of ubiquitin-specific proteases (USPs) is a novel and promising direction in the development of molecularly targeted therapies in oncology. The aim of the present study was to examine whether Degrasyn could be a potential therapeutic agent against bladder cancer (BC). Also, we aimed to determine whether Degrasyn is more effective in terms of anti-cancer activity compared to the non-selective DUB inhibitor PR-619. To facilitate the translational value of the obtained results, our experiments were performed using both human and canine in vitro models of BC. Methods: Human T24 (urothelial grade III BC) and SV-HUC-1 (non-tumorigenic urothelial cell line), as well as canine K9TCC-PU-NK and RDSVS-TCC1 (both derived from invasive grade III urothelial bladder tumors) cell lines, were used in the present study. Cell proliferation was determined using the MTT assay and Ki-67 proliferation assay, and the level of apoptosis induced by Degrasyn and PR-619 was evaluated by Annexin V-FITC staining and caspase 3/7 activation assay. Western blot was used to assess DNA damage and key proteins involved in apoptosis. Results: Degrasyn inhibited the proliferation of all BC cell lines in a concentration- and time-dependent manner. Lower concentrations of Degrasyn were more potent against human and canine BC cell lines compared to PR-619. Degrasyn induced caspase-dependent apoptosis and triggered DNA damage. PR-619 did not show a significant pro-apoptotic effect. Conclusions: Our results demonstrate that Degrasyn significantly impairs the growth of in vitro models of human and canine BC. Selective USP inhibition with Degrasyn seems to be more effective in reducing BC cell proliferation and inducing apoptosis and DNA damage than non-selective USP inhibition with PR-619.

## 1. Introduction

Bladder cancer (BC) is the most common malignancy of the urinary system, with approximately 500,000 individuals diagnosed annually worldwide [1]. BC is a heterogeneous disease associated with various clinical outcomes. It may be divided into two main staging subgroups: non-muscle-invasive bladder cancer (NMIBC) and muscle-invasive bladder cancer (MIBC) [2]. NMIBC includes lesions limited to the mucosal or submucosal layers of the bladder wall, whereas MIBC comprises tumors infiltrating and/or exceeding the lamina muscularis propria of the bladder wall [2].

The majority of NMIBCs can be managed with total curative intent, while the treatment of MIBCs is very challenging and multimodal. Despite the standard treatment (involving radical cystectomy with pelvic lymph node dissection), almost 50% of patients with MIBC will eventually develop metastatic disease, which is associated with poor survival rates [3]. The systemic management of advanced or metastatic MIBC primarily consists of platinum-based chemotherapy; however, chemotherapy provides an approximately 50% response rate with only a 20% 5-year overall survival rate [3]. One of the most important discoveries in recent years, facilitating the treatment of advanced or metastatic MIBC, has been the introduction and approval of drugs targeting immune checkpoints: programmed death receptor 1 (PD-1) and programmed death receptor 1 ligand (PD-L1) inhibitors. Nevertheless, despite their efficacy, which has been proven in an increasing number of clinical trials, only a small proportion of patients clearly benefit from such treatment (which is mainly dependent on PD-1/PD-L1 expression levels) [4]. Due to the above-mentioned limitations of current systemic MIBC therapies, great emphasis is constantly placed on the investigation of novel therapeutic targets and drugs.

Since its discovery almost 40 years ago, the ubiquitin-proteasome system (UPS) has been the subject of increased research interest, as protein ubiquitination and related post-translational modifications regulate diverse aspects of cellular biology. Generally, the process of ubiquitination requires three different enzymes: E1 (ubiquitin-activating enzyme), E2 (ubiquitin-conjugating enzyme), and E3 (ubiquitin ligase) [5]. It can be reversed by deubiquitinating enzymes (DUBs), a group of over 100 enzymes divided into five families: ubiquitin-specific proteases (USPs); ovarian tumor proteases (OTUs); ubiquitin C-terminal hydrolases (UCHs); Machado–Joseph disease protein domain proteases (MJDs); and JAMM motif proteases [5]. Particular DUBs catalyze the removal of ubiquitin particles from different ubiquitinated proteins, which leads to the regulation of the stability and activity of the target particles [5,6]. This process is crucial for controlling various cellular pathways, such as DNA repair, gene expression, protein localization, kinase activation, cell cycle progression, or cell apoptosis [6]. The growing body of evidence suggests that dysregulation of the DUBs system can contribute to the development of a variety of cancers, including BC [6].

Being the largest family among DUBs with over 50 members, USPs have attracted major attention as potential molecular targets in cancer therapy. To date, pharmacological studies have identified several novel DUB inhibitors for potential clinical use, mainly non-selective multiple DUBs/USPs inhibitors [7]. In the BC setting, a few preclinical studies demonstrated that the non-selective pan-DUB/USP inhibitor PR-619 appears to be a potential therapeutic agent, especially in combination with chemotherapeutics [8,9]. Recently, more selective DUBs/USPs inhibitors, such as Degrasyn (also known as WP1130), have been investigated in some types of cancer with promising results. Degrasyn is a small molecule compound identified as a selective USP (USP5, USP9X, and USP14) inhibitor. The effectiveness of Degrasyn against BC has not been investigated to date.

BC is not only a significant health issue in humans, but it also occurs in companion animals, including dogs (representing 1–2% of all canine tumors) [10]. With estimates that 4–6 million pet dogs develop cancer in the United States annually, this equates to more than 60,000 cases of BC in dogs each year. In contrast to humans, the majority of dogs (90% of cases) are diagnosed with advanced or metastatic MIBC, which is frequently characterized by an aggressive clinical course and chemotherapy resistance [10]. Therefore, the generally poor survival outcomes in dogs with BC implicate an urgent need for the development of novel treatment compounds simultaneously with human medicine. Successful findings from research using both human and canine models can be potentially translated across the species, accelerating the development of novel therapeutic agents effective against BC in either human or veterinary medicine.

As the inhibition of DUBs/USPs functions is a novel and promising direction in the development of molecularly targeted therapies in oncology, the main aim of the present study was to examine whether Degrasyn could be a potential therapeutic agent effective against bladder cancer (BC). Also, we aimed to determine whether Degrasyn is more effective in terms of anti-cancer activity compared to the non-selective DUB inhibitor PR-619. To facilitate the translational value of the obtained results, our experiments were performed using both human and canine in vitro models based on multidisciplinary collaboration.

## 2. Materials and Methods

### 2.1. Cell Lines and Cell Culture

Human T24 (urothelial grade III BC) and SV-HUC-1 (immortalized and non-tumorigenic urothelial cells) cell lines were used in the present study. They were acquired from the American Type Culture Collection (ATCC, Rockville, MD, USA). T24 and SV-HUC-1 cell lines were maintained in McCoy’s 5a Modified Medium and F-12K Medium (ATCC, Rockville, MD, USA), respectively. As a representative in vitro model of canine BC, K9TCC-PU-NK and RDSVS-TCC1 cell lines (both derived from invasive grade III urothelial bladder tumors) were also involved in this study. K9TCC-PU-NK and RDSVS-TCC1 cell lines were generous gifts from Deepika Dhawan (Purdue University College of Veterinary Medicine, USA) and Maciej Parys (Royal (Dick) School of Veterinary Studies and The Roslin Institute, University of Edinburgh, Scotland), respectively. Both canine BC cell lines were cultured in Roswell Park Memorial Institute (RPMI) 1640 Medium (Institute of Immunology and Experimental Therapy, Polish Academy of Sciences, Wrocław, Poland).

All culture media used in our study were supplemented with 2 mM L-glutamine, 100 U/mL penicillin, 100 µg/mL streptomycin (Sigma-Aldrich, Steinheim, Germany), and 10% heat-inactivated fetal bovine serum (FBS, Gibco, Grand Island, NY, USA). All cell cultures were performed at 37 °C in a humidified atmosphere containing 5% CO_2_.

### 2.2. Chemicals and Reagents

Degrasyn (WP1130) and PR-619 were purchased from Selleck Chemicals (Houston, TX, USA). FBS, L-glutamine, penicillin and streptomycin solution, 0.25% trypsin EDTA, trypan blue, poly-L-lysine, ribonuclease A, propidium iodide (PI), RIPA buffer, SigmaFAST Protease Inhibitor Cocktail, and TBST buffer were acquired from Sigma-Aldrich (Steinheim, Germany). Annexin V-FITC was obtained from Immunostep (Salamanca, Spain). The CellEvent^®^ Caspase3/7 Green Flow Cytometry Assay by ThermoFisher (Waltham, MA, USA) was used. Anti-USP5 (sc-390943), anti-USP9X/Y (sc-365353), anti-USP14 (sc-515812), anti-Bcl-2 (sc-7382), and anti-β actin (sc-47778) antibodies were purchased from Santa Cruz Biotechnology (Santa Cruz, CA, USA), whereas the anti-Bcl-xl (#2764) antibody was purchased from Cell Signaling (Cell Signaling Technology, Danvers, MA, USA). The anti-γH2A.X (ab26350) antibody was from Abcam (Cambridge, UK), and anti-mouse/HRP and anti-rabbit/HRP antibodies were bought from Dako (Glostrup, Denmark).

### 2.3. Cell Proliferation Assay

Cell proliferation was determined using the 3-(4,5-dimethylthiazol-2-yl)-2,5-diphenyltetrazolium bromide (MTT) assay (Sigma-Aldrich, Steinheim, Germany). Cells were plated at a density of 1 × 10^5^ cells per mL in 96-well plates (Thermo Fisher Scientific, Roskilde, Denmark) and allowed to attach overnight. Subsequently, Degrasyn and PR-619 were added in increasing concentrations (0.6, 1.25, 2.5, 5, 10, and 20 μM). After incubation for 24, 48, and 72 h with either the compounds or in the medium alone, 20 μL of MTT solution (5 mg/mL) was added to each well. After an additional 4 h, 80 μL of the lysis buffer (225 mL DMF, 67.5 g SDS, and 275 mL distilled water) was added to each well. After dissolving the content, the absorbance was measured at a reference wavelength of 570 nm using a spectrophotometric microplate reader Spark (Tecan). The values were obtained from 3 independent experiments (different plates, different days). Based on the results, the concentration that inhibited 50% of the cell proliferation (IC_50_) was calculated for all incubation intervals.

### 2.4. Ki-67 Flow Cytometry Proliferation Assay

For the cell proliferation assay, cells were plated in 100 mm cell culture plates. When the cultures reached 60% confluence, the media were discarded from the wells and replaced with culture media only or media with added compounds (Degrasyn, PR-619) with increasing concentrations for 24 h. The growth medium was then removed from the wells, the cells were rinsed twice with PBS, and 0.25% trypsin-EDTA was added to detach the cells. After 5 min of incubation at 37 °C and 5% CO_2_, the fresh culture medium was added to inactivate the trypsin, and cells were collected in flow cytometry tubes. Cells were then labeled using the Ki-67 Proliferation Kit BD Pharmingen(BD Biosciences) according to the manufacturer’s instructions and analyzed using a flow cytometer (Cytoflex, Beckman Coulter). Unstained control cells were used for gating to determine the percentage of the proliferation of the Ki-67-positive cells in the samples. Percentages of proliferating (Ki-67 positive) cells were used to calculate the means ± SD for each cell line. The presented results were obtained from three independent experiments.

### 2.5. Western Blot Analysis

Western blot was performed to assess the expression of USP5, USP9X, and USP14 in analyzed cell lines. After 24 h of incubation with increasing concentrations of Degrasyn and PR-619, the expression of proteins involved in apoptosis (Bcl-2 and Bcl-xl) and a marker of DNA damage (γH2A.X) was also evaluated. Briefly, cells were rinsed with cold phosphate-buffered saline (PBS), suspended in lysis buffer (50 mM Tris-HCl pH 7.5, 100 mM NaCl, 1% NP-40, protease inhibitors set), and incubated for 20 min on ice. Subsequently, after centrifuging at 10,000 rpm at 4 °C for 12 min, sodium dodecyl sulfate (SDS) sample buffer was added to the suspensions to clear the supernatants. After boiling for 5 min at 95 °C, the samples were subjected to SDS-polyacrylamide gel electrophoresis on 8–15% gel. For USP5, USP9X, USP14, γH2A.X, Bcl-2, and Bcl-xl expression analysis, transfers onto nitrocellulose membranes were performed using a Semidry Transfer Cell or wet transfer method (Bio-Rad, Hercules, CA, USA). After transfer, the membranes were blocked with 1% casein in TBS (or 3% BSA for γH2Ax) at room temperature for 1 h. After blocking, the membranes were incubated overnight at 4 °C with primary antibodies: anti-USP5 (dilution 1:500), anti-USP9X/Y (dilution 1:500), anti-USP14 (dilution 1:500), anti-γH2A.X (dilution 1:1000), anti-Bcl-2 (dilution 1:1000), anti-Bcl-xl (dilution 1:1000), and murine monoclonal anti-β-actin antibody (dilution 1:2000). The membranes were then incubated with the secondary antibody for 1 h at room temperature. Membrane visualizations were performed using ChemiDoc Touch Instruments (Bio-Rad, Hercules, CA, USA).

### 2.6. Quantification of Apoptosis

To assess the level of apoptosis induced by Degrasyn and PR-619, cells were seeded at a density of 1 × 10^5^/mL in 96-well plates (TPP, Trasadingen, Switzerland) and incubated for 24 h with increasing concentrations of compounds. Cells were then collected, suspended in a binding buffer, and stained with Annexin V-FITC for 10 min at room temperature. Subsequently, PI was added, and flow cytometric analysis was immediately performed using a flow cytometers (FACS Calibur, Becton Dickinson, Biosciences, San Jose, CA, USA; CytoFlex, Beckman Coulter, CA, USA). CellQuest 3.lf. (Becton Dickinson, San Jose, CA, USA) and CytExpert 2.4 (Beckman Coulter, CA, USA) software were used for data analysis.

Subsequently, to assess caspases 3/7 activation, the cells were incubated with medium alone or with increasing concentrations of compounds (Degrasyn, PR-619) in conditions similar to the previous test. Following cell collection, the cells were twice PBS-washed and stained in accordance with the manufacturer’s instructions. For active caspase 3/7 detection, CellEventCaspase-3/7 Green Detection Reagent Solution was added to the samples. The cells were then suspended 1 more time in the wash buffer after being washed twice. For flow cytometric analysis, a flow cytometers (FACS Calibur, Becton Dickinson, Biosciences, San Jose, CA, USA; CytoFlex, Beckman Coulter, CA, USA) were used, and CellQuest 3.lf (Becton Dickinson, San Jose, CA, USA) and CytExpert 2.4 (Beckman Coulter, CA, USA) software were used for data analysis. The presented results were obtained from three independent experiments.

### 2.7. Statistical Analysis

Statistical analyses were performed using the GraphPad Prism 9 software (GraphPad Software, San Diego, CA, USA). All data were presented as means with standard deviations (SD). Data from the 2 groups were analyzed by a 2-tailed Student’s t-test, and data from multiple groups were analyzed using a 1-way ANOVA followed by Tukey’s posthoc test. Results were considered significant at *p* < 0.05.

## 3. Results

### 3.1. Expression of USP5, USP9X, and USP14 in Human and Canine Bladder Cancer Cells

At the first step of the research, we analyzed USP5, USP9X, and USP14 expression levels in selected cell lines, as they are targets of the inhibitory effect of Degrasyn. As illustrated in Figure 1, both human cell lines (T24 and SV-HUC-1) were characterized by higher expression levels of the USP5 protein compared to canine BC cell lines (K9TCC-PU-NK and RDSVS-TCC1). Compared to control non-tumorigenic urothelial cells (SV-HUC-1 cell line), higher expression of USP5 protein was observed in the T24 cell line (representing a human BC cell line). Similar expression levels of USP9X and USP14 proteins were observed in either human or canine cell lines.

### 3.2. Determination and Comparison of the Anti-Proliferative Effects of Degrasyn and PR-619 on Bladder Cancer Cells

Subsequently, in order to evaluate the hypothesis of the antiproliferative activity of Degrasyn against BC cells, we performed an MTT assay. The MTT assay demonstrated that Degrasyn inhibited the proliferation of all cell lines used in the present study in a concentration- and time-dependent manner (Table 1). The number of actively dividing cells clearly decreased between 24 and 48 h of incubation, but the difference between 48 h and 72 h was marginal. Relative to 24 h of incubation time, we observed significantly less cytotoxicity of Degrasyn on the human control SV-HUC-1 cell line compared to the T24 cell line (IC_50_ values 2.33 μM and 3.07 μM, respectively, *p* < 0.05). However, the IC_50_ values did not significantly differ for longer incubation times (Table 1). For all incubation times, the human BC line (T24) was more sensitive to the anti-proliferative effect of Degrasyn compared to the canine BC cell lines (K9TCC-PU-NK and RDSVS-TCC1). 

Furthermore, we compared the antiproliferative activity of Degrasyn and PR-619. The MTT assay confirmed the antiproliferative activity of PR-619 against all analyzed cell lines. The effect was both concentration- and time-dependent (Table 2). The IC_50_ of Degrasyn was significantly lower than the IC_50_ of PR-619 in corresponding time intervals (Figure 2). Lower concentrations of Degrasyn (1.25–10 µM) were more potent toward human and canine cell lines compared to PR-619 (Figure 3, Appendix A). Nevertheless, the highest concentrations of both Degrasyn and PR-619 used in this study killed almost 100% of cells in all analyzed cell lines.

### 3.3. Ki-67 Flow Cytometry Proliferation Assay

The percentage of Ki-67 positive cells (as compared to the control) was significantly decreased after incubation with 2.5–5 µM Degrasyn and 5–10 µM PR-619 in the T24 cell line (*p* < 0.05) (Figure 4, Appendix A). For other cell lines, the percentage of Ki-67 positive cells (as compared to control) was significantly decreased (*p* < 0.05) after incubation only with the highest concentrations of compounds (5 µM Degrasyn and 10 µM PR-619) (Figure 4, Appendix A). Generally, the proportion of Ki-67 positive cells was significantly lower (*p* < 0.05) in SV-HUC-1 and canine BC cell lines than in the T24 cell line for all used concentrations of Degrasyn and PR-619 (*p* < 0.05).

### 3.4. Determination and Comparison of Pro-Apoptotic Effect of Degrasyn and PR-619 on Bladder Cancer Cells

In the next step, we determined whether the inhibition of cell proliferation by Degrasyn demonstrated in the MTT assay was potentially related to the induction of apoptosis. For this purpose, Annexin V-FITC and PI staining were performed. After 24 h of incubation with Degrasyn at increasing concentrations (up to approximately the IC_50_ value for particular cell lines), we detected a significant increase in the number of apoptotic cells (stained only with Annexin V-FITC for early apoptotic cells and double stained with Annexin-FITC and PI for late apoptotic cells) in all cell lines except the K9TCC-PU-NK cell line (Figure 5). The effect of Degrasyn on apoptosis level was concentration-dependent, and the percentage of apoptotic cells was the highest in T24 and RDSVS-TCC1 cell lines (for the highest used concentrations representing an approximate IC_50_ value) (Figure 6). In contrast, PR-619 did not show a significant pro-apoptotic effect in the analyzed cell lines. For the highest used concentrations representing approximate IC_50_ values of both compounds, we detected a lower increase in the number of apoptotic cells (stained only with Annexin V-FITC for early apoptotic cells and double stained with Annexin-FITC and PI for late apoptotic cells) after 24 h of incubation with PR-619 compared to Degrasyn in all analyzed cell lines (Figure 5). The percentages of early and late apoptotic cells after 24 h of incubation with 2.5 μM Degrasyn and 5 μM PR-619 were: 24.1 ± 2.8% and 14.8 ± 5.8% in the SV-HUC-1 cell line, respectively; 32.2 ± 9.2% and 9.8 ± 1.5% in the T24 cell line, respectively; 11.8 ± 3.4% and 8.4 ± 0.8% in the K9TCC-PU-NK cell line, respectively; 42.4 ± 2.3% and 17.2 ± 6.5% in the RDSVS-TCC1 cell line, respectively.

To further assess the role of caspases in the apoptosis induced by Degrasyn and PR-619, we evaluated the activation of caspase 3/7. The results are presented in Figure 7 and Figure 8. In the SV-HUC-1 cell line, the activation of caspase 3/7 was significantly higher compared to the control (medium only) only after incubation with the highest used concentration of Degrasyn (2.5 μM). For the T24 cell line, the activation of caspase 3/7 was significantly higher compared to the control (medium only) after incubation with 1.25 μM and 2.5 μM Degrasyn). In both canine BC cell lines (K9TCC-PU-NK and RDSVS-TCC1), the activation of caspase 3/7 was significantly higher after incubation with 2.5 μM and 5 μM Degrasyn as compared to control (medium only). Generally, activation of caspase 3/7 was less pronounced in canine BC cell lines than in the T24 cell line after incubation with Degrasyn. Specifically, the number of cells with active caspase 3/7 after the incubation with the highest concentrations of Degrasyn reached 22.4 ± 5.4% in the SV-HUC-1 cell line, 36.6 ± 0.7% in the T24 cell line, 11.9 ± 1.8% in the K9TCC-PU-NK cell line, and 22.5 ± 4.3% in the RDSVS-TCC1 cell line. After the incubation with the highest concentration of PR-619, the percentage of cells with active caspase 3/7 was 19.6 ± 1.8% in the SV-HUC-1 cell line, 16.1 ± 4.9% in the T24 cell line, 13.3 ± 3.3% in the K9TCC-PU-NK cell line, and 16.2 ± 7.4% in RDSVS-TCC1 cell line. 

### 3.5. Evaluation of USP5 Expression in T24 Cells after 24 h of Incubation with Degrasyn and PR-619

To determine the impact of Degrasyn and PR-619 on USP5 protein expression level, a Western blot was performed after a 24 h incubation with both compounds in the human T24 cell line (characterized by the highest expression of USP5 protein among analyzed cell lines). As shown in Figure 9, Degrasyn clearly decreased the USP5 protein expression; however, no visible effect on USP5 expression was observed after 24 h of incubation with PR-619.

### 3.6. Evaluation of DNA Damage in Bladder Cancer Cells Induced by Degrasyn and PR-619

Because γH2AX is a recognized marker of DNA damage, we measured its levels after 24 h of incubation with increasing concentrations of Degrasyn and PR-619. Figure 10 shows that treatment of all used BC cell lines with Degrasyn increased the amount of γH2AX at higher concentrations of the drug. In the T24 cell line, a high expression of γH2AX was observed after incubation with 2.5 µM and 5 µM Degrasyn. In canine BC cell lines, a high expression of γH2AX was observed only after incubation with 5 µM Degrasyn. Incubation with PR-619 resulted in only a slight visible increase in the amount of γH2AX after incubation with the highest concentrations of the drug.

### 3.7. Evaluation of Proteins Involved in Apoptosis Induced by Degrasyn and PR-619

The involvement of key proteins in cell apoptosis after Degrasyn and PR-619 treatment was analyzed using Western blot (Figure 11). No expression of the anti-apoptotic protein Bcl-2 was observed in the SV-HUC-1 cell line, which could be explained by the fact that it is a non-cancerous cell line. The basal expression level of Bcl-2 and Bcl-xl was comparable between human and canine BC cell lines (T24, K9TCC-PU-NK, and RDSVS-TCC-1). Expression of Bcl-2 protein was slightly decreased after incubation with the highest concentrations of drugs (5 μM Degrasyn and 10 μM PR-619 in all BC cell lines. In all cell lines, Bcl-xl expression was slightly reduced after incubation with a 5 μM concentration of Degrasyn. No clearly visible changes in the expression levels of Bcl-xl were noticed in the analyzed cell lines after incubation with increasing concentrations of PR-619.

## 4. Discussion

Given that DUBs and USPs target proteins contain a large number of cell homeostasis regulators as well as products of known oncogenes or tumor suppressor genes, DUBs and USPs inhibition has emerged as an appealing and promising research direction for the development of novel cancer therapies [11]. As Degrasyn (a selective inhibitor of USP5, USP9X, and USP14) has shown an inhibitory effect on some cancer types, our aim was to determine whether Degrasyn could be a potential therapeutic agent in BC. We demonstrated that Degrasyn is an effective agent against human and canine BC cells. Degrasyn exhibited anti-proliferative, pro-apoptotic, and DNA-damage effects on BC cells that were more pronounced compared to the non-selective DUB inhibitor PR-619.

Initially, we confirmed the expression of USP5, USP9X, and USP14 proteins in selected BC cell lines, which provided the rationale for the utilization of Degrasyn in a BC setting. We observed the overexpression of USP5 protein in a human BC cell line (T24), as compared to normal urothelial cells. Accumulating data suggest that overexpression of USP5 contributes to tumorigenesis in various cancer types through deubiquitinating and stabilizing oncoproteins, such as p53 in melanoma, c-Maf in multiple myeloma, Slug in hepatocellular carcinoma, or histone deacetylase 2 (HDAC2) in ovarian cancer [12]. We evaluated the antiproliferative effect of Degrasyn using the MTT assay. The present study demonstrated that, from all investigated cell lines, the T24 cell line (human BC cells) was most sensitive to the anti-proliferative effect of Degrasyn. This could be potentially related to the observed overexpression of USP5 protein in human BC cells. Some previous studies demonstrated that the anti-cancer activity of Degrasyn could be mainly related to the USP5 inhibitory effect. Li et al. found that Degrasyn suppressed proliferation, reduced colony formation, and inhibited metastasis in pancreatic cancer cells through the USP5-WT1E-cadherin axis [13]. In another paper by Zhang et al., the authors showed that Degrasyn displayed marked activity in downregulating CCND1 protein and suppressing non-small cell lung carcinoma proliferation, which was consistent with the inhibitory effect elicited by the knockdown of USP5 expression [14]. 

Having analyzed the impact of Degrasyn on cell proliferation, we investigated if Degrasyn may cause any particular type of cell death. We demonstrated that Degrasyn might cause apoptosis in BC cells. Our results were consistent with previous reports on multiple myomas [15]. Additionally, comparing the results of the MTT assay with the analysis of apoptosis level at the same concentration and time of incubation, it can be concluded that at the lower concentrations of the compound, it had a mainly anti-proliferative effect, and with its increase, the reduction of cell viability was increasingly related to apoptosis. Anti-apoptotic effect of PR-619 was less pronounced. Prior research has shown that inhibition of USP can induce cell death via caspase-dependent apoptosis [16]. The activation of effector caspases 3/7, which are proteases responsible for protein cleavage, DNA condensation, and other apoptotic markers, is one of the critical steps during the process [17]. We were able to identify the activation of caspase 3/7 in analyzed cell lines treated with Degrasyn and proved that caspase activation was essential in the apoptosis induced by the agent. In addition, we observed decreased expression of Bcl-2 protein after incubation with the highest concentrations of Degrasyn (5 μM Degrasyn) in all BC cell lines. Also, Bcl-xl expression was slightly reduced after incubation with a 5 μM concentration of Degrasyn. As it has been proven that substrates for Degrasyn (e.g., USP9X) might regulate the Bcl-2 family axis [18], their inhibition may contribute to the anti-cancer effect of Degrasyn.

According to several studies, many USPs are involved in DNA damage repair pathways (DDR) [19,20]. Various ubiquitination pathways have distinct molecular structural characteristics and biological activities, including ubiquitin chains that are critical to the DDR process. Nakajima et al. demonstrated that USP5 is involved in the elimination of the ubiquitin signal from damaged sites and is required for efficient DNA double-strand break (DSB) repair [20]. In another study, Zheng et al. demonstrated that inhibition of USP5 in non-small cell lung cancer can cause DNA damage while modulating the transcriptional activity of p53 [19]. In our study, DNA damage (significant upregulation of the appearance of γH2A.X expression) was found to be an effect of Degrasyn action in analyzed BC cell lines. As the majority of invasive BCs have alterations in the p53 pathway, a p53-dependent mechanism of Degrasyn action might be related to DNA damage in BC cells; however, the detailed mechanism has to be further evaluated.

While there is great interest in DUBs/USPs as a way to provide a high-precision mechanism for the degradation of their substrate proteins, the number of selective compounds is limited to date [7,21]. One of the most difficult challenges in developing inhibitors selective for a specific DUB or USP is the similarity between DUB family members (especially USPs), which results in compounds with poor selectivity profiles and limits their utility in elucidating DUB or USP function [7,11,21]. A majority of the reported DUBs/USPs inhibitors exhibited weak inhibitory activity (double-digit micromolar range) [7,11,21,22]. Recently, the non-selective multiple DUBs/USPs inhibitor PR-619 was investigated as a potential therapeutic agent in BC. Hsu et al. revealed that PR-619 enhanced cisplatin-induced cytotoxicity and alleviated cisplatin resistance in the cisplatin-resistant T24/R (cisplatin-resistant BC cells) cell line and concurrently suppressed c-Myc expression [9]. Similarly, Kuo et al. demonstrated that PR-619 could inhibit the BC cells’ growth alone as well as effectively enhance the cisplatin-induced antitumor effect via concurrent suppression of the Bcl-2 level [8]. In the present study, we observed a significantly higher anti-proliferative effect of Degrasyn than PR-619 at corresponding concentrations. Further, we demonstrated a lower apoptosis induction in BC cells after 24 h of incubation with PR-619 compared to Degrasyn. Additionally, the effect on DNA damage induction was also less pronounced. Thus, our results are in line with existing evidence regarding the potential superiority of selective DUB/USP inhibitors in terms of anti-cancer activity as compared to non-selective inhibitors [7,11,21,22].

In our study, we also analyzed the anti-cancer activity of Degrasyn and PR-619 on two canine BC cell lines (representing in vitro models of invasive BC in dogs). Generally, canine MIBC shares many similarities with human MIBC, including protein and gene homology, pathophysiological mechanisms of cancer initiation and progression, drug targets, drug resistance, and potential prognostic and diagnostic biomarkers [23]. The multidisciplinary collaboration and utilization of canine BC cell lines in our study were justified by providing translational data that could be implemented not only in human oncology but also in veterinary oncology. Although the observed anti-proliferative effect was higher in human BC cells, for the first time, we demonstrated that both Degrasyn and PR-619 could be potential therapeutic agents in canine BC. In our study, differences between human and canine cell lines could be explained by different USP5 expressions and caspase 3/7 activation; however, several other unknown factors may contribute to them and need to be elucidated in further studies.

## 5. Conclusions

Our results demonstrate that Degrasyn significantly impairs the growth of in vitro models of human and canine BC. Selective USP inhibition with Degrasyn seems to be more effective in reducing BC cell proliferation and inducing apoptosis and DNA damage than non-selective USP inhibition with PR-619.

## Figures and Tables

**Figure 1 biomedicines-11-00759-f001:**
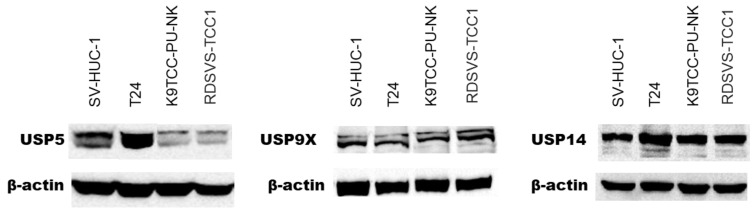
Representative graphical results of Western blot analyses showing expression of USP5, USP9X and USP14 in tested cell lines. USP = ubiquitin-specific protease.

**Figure 2 biomedicines-11-00759-f002:**
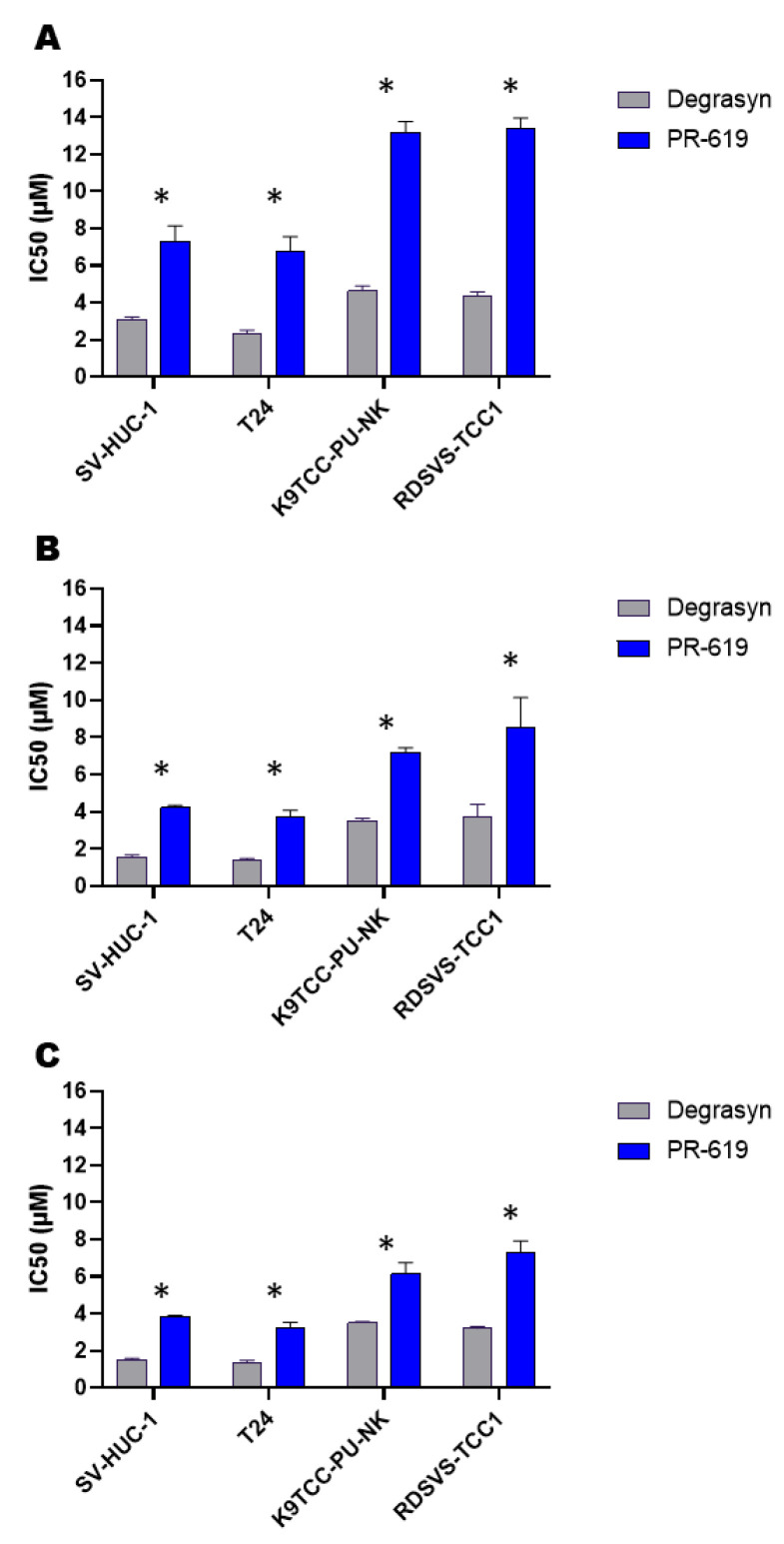
Comparison of Degrasyn and PR-619 half maximal inhibitory concentrations (IC_50_) values after (**A**) 24 h incubation; (**B**) 48 h incubation; (**C**) 72 h incubation. The values are presented as means ± standard deviations (SD) of three independent experiments. * Difference between Degrasyn and PR-619 considered statistically significant (*p <* 0.05).

**Figure 3 biomedicines-11-00759-f003:**
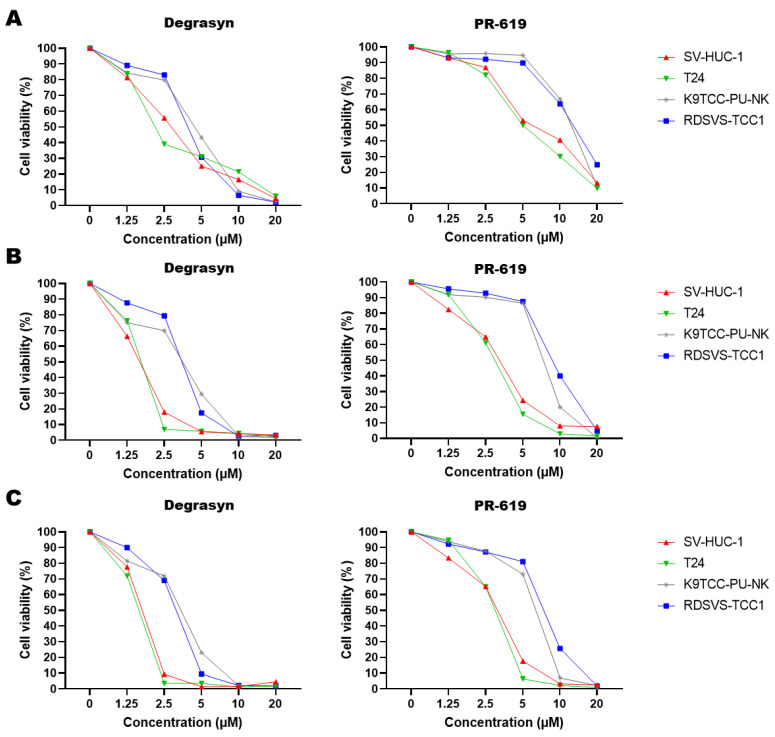
Analysis of Degrasyn and PR-619 influence on the cell proliferation of SV-HUC-1, T24, K9TCC-PU-NK, and RDSVS-TCC1 cell lines (MTT assay) after 24 (**A**), 48 (**B**), and 72 (**C**) hours with increasing concentrations of the drugs (data for 1.25, 2.5, 5, 10, and 20 μM). Values are presented as means ± standard deviations (SD) of three independent experiments.

**Figure 4 biomedicines-11-00759-f004:**
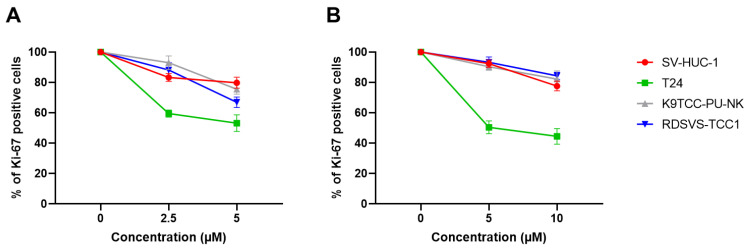
Ki-67 flow cytometry proliferation assay. Percentages of proliferating (Ki-67 positive) cells in reference to control (medium only) after the incubation with increasing concentrations of (**A**) Degrasyn and (**B**) PR-619. Values are presented as means ± standard deviations (SD) of three independent experiments.

**Figure 5 biomedicines-11-00759-f005:**
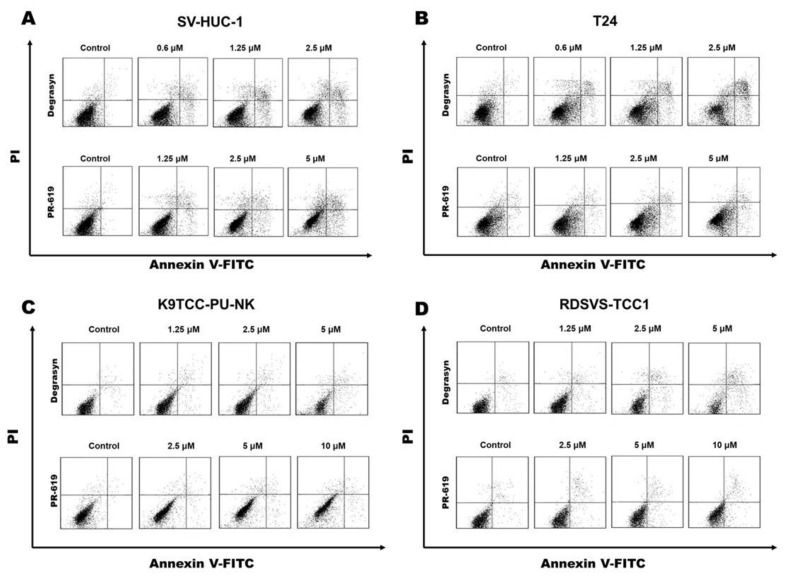
Evaluation of apoptosis induced by Degrasyn and PR-619. Representative dot-plots showing Annexin V-FITC/PI staining of (**A**) SV-HUC-1, (**B**) T24, (**C**) K9TCC-PU-NK, and (**D**) RDSVS-TCC1 cells after 24 h exposure to increasing concentrations of Degrasyn and PR-619.

**Figure 6 biomedicines-11-00759-f006:**
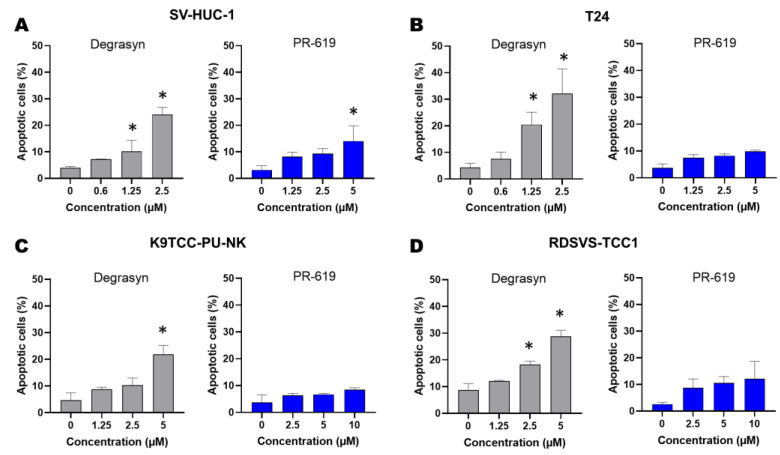
Evaluation of apoptosis induced by Degrasyn and PR-619. Percentage of apoptotic cells after 24 h exposure to increasing concentrations of Degrasyn and PR-619 in (**A**) SV-HUC-1, (**B**) T24, (**C**) K9TCC-PU-NK, and (**D**) RDSVS-TCC cell lines (Annexin V-FITC/PI staining). Values are expressed as the means ± standard deviations (SD) of three independent experiments. * Considered significant in comparison to control (*p* < 0.05).

**Figure 7 biomedicines-11-00759-f007:**
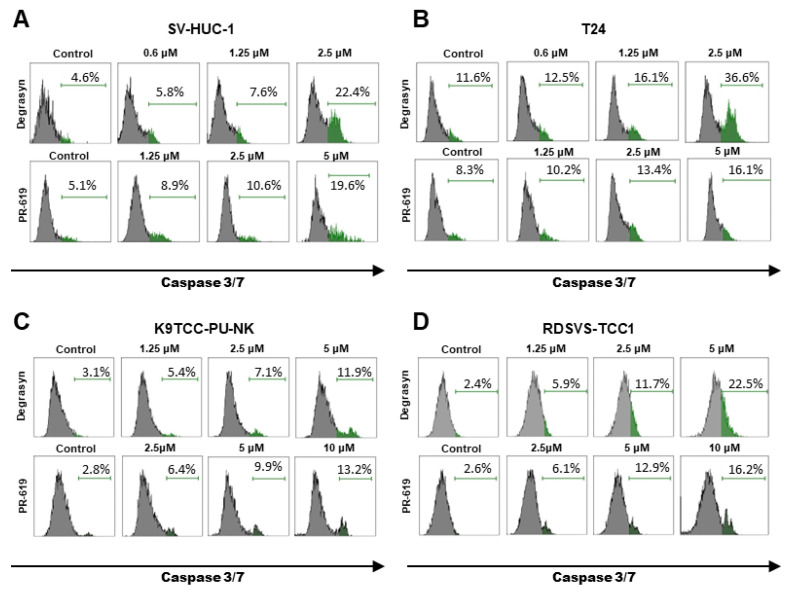
Evaluation of caspase 3/7 activation induced by Degrasyn and PR-619. Representative histograms showing the percentage of cells stained with CellEvent^®^Caspase-3/7 Green Detection Reagen after 24 h exposure to increasing concentrations of Degrasyn and PR-619 in (**A**) SV-HUC-1, (**B**) T24, (**C**) K9TCC-PU-NK, and (**D**) RDSVS-TCC cell lines.

**Figure 8 biomedicines-11-00759-f008:**
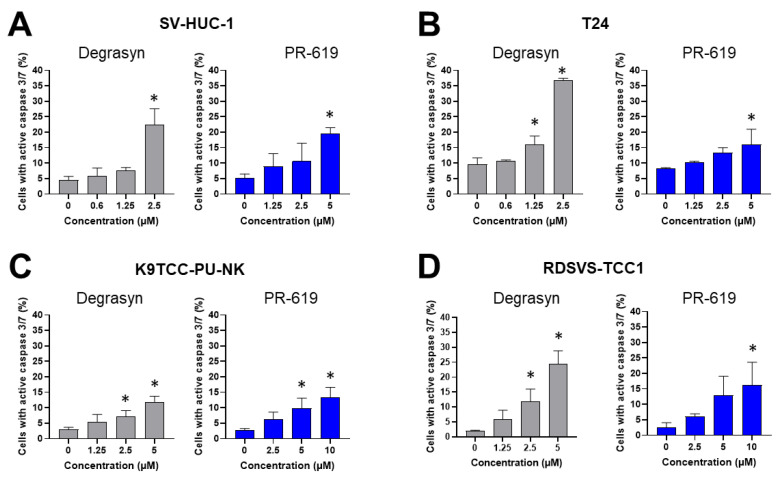
Evaluation of caspase 3/7 activation induced by Degrasyn and PR-619. Percentage of cells with active caspase 3/7 after 24 h exposure to increasing concentrations of Degrasyn and PR-619 in: (**A**) SV-HUC-1, (**B**) T24, (**C**) K9TCC-PU-NK, and (**D**) RDSVS-TCC cell lines. Values are expressed as the means ± standard deviations (SD) of three independent experiments. * Considered significant in comparison to control (*p* < 0.05).

**Figure 9 biomedicines-11-00759-f009:**
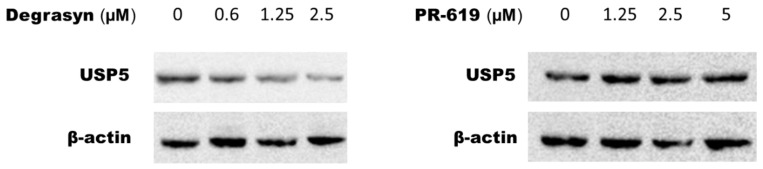
Western blot analysis for USP5 expression in T24 cells after 24 h of incubation with increasing concentrations of Degrasyn (0.6, 1.25, and 2.5 μM) and PR-619 (1.25, 2.5, and 5 μM). USP = ubiquitin-specific protease.

**Figure 10 biomedicines-11-00759-f010:**
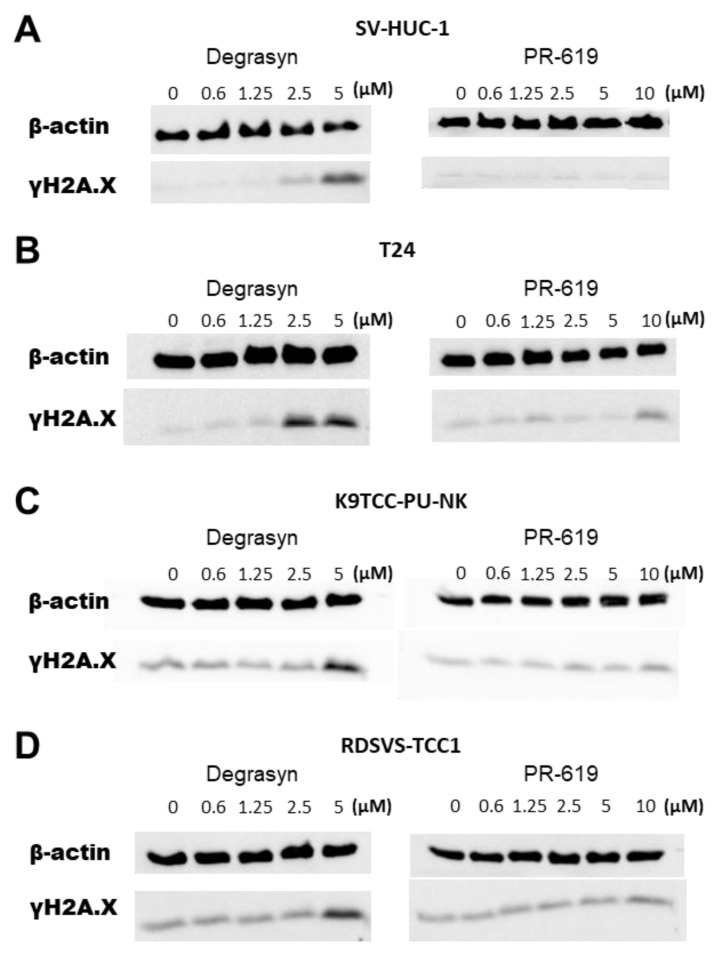
Evaluation of DNA damage generated by Degrasyn and PR-619. Representative blot of γH2A.X expression in (**A**) SV-HUC-1, (**B**) T24, (**C**) K9TCC-PU-NK, and (**D**) RDSVS-TCC cells after 24 h of treatment with increasing concentrations of Degrayn and PR-619. Actin serves as a loading control.

**Figure 11 biomedicines-11-00759-f011:**
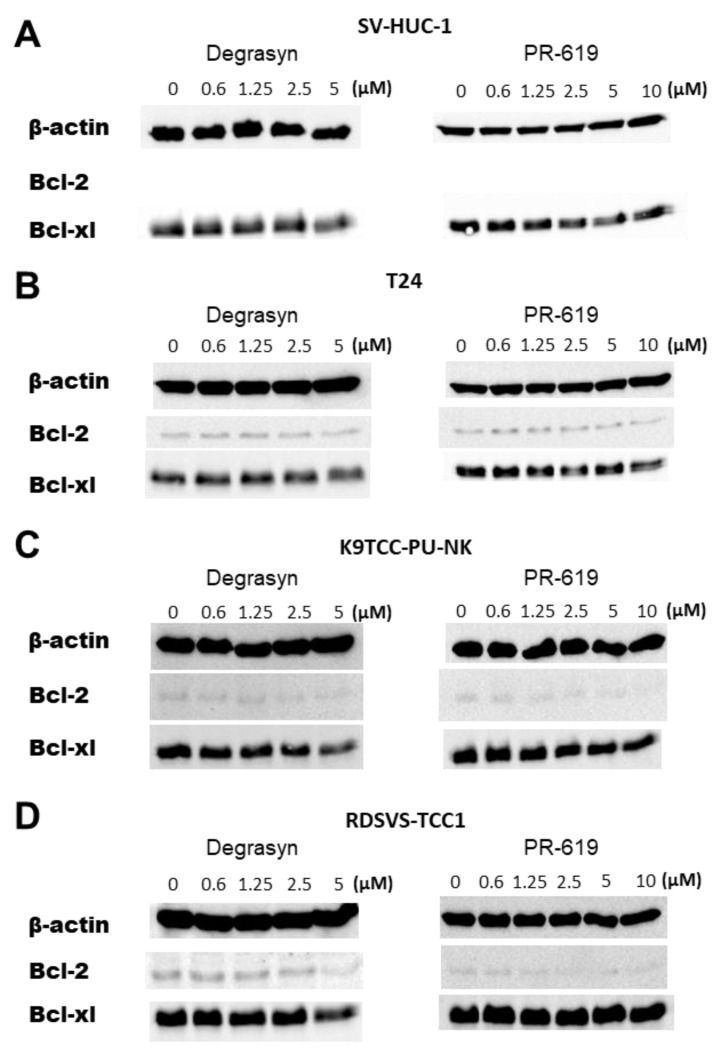
Evaluation of proteins involved in apoptosis induced by Degrasyn and PR-619. Representative blot of Bcl-2 and Bcl-xl expression in (**A**) SV-HUC-1, (**B**) T24, (**C**) K9TCC-PU-NK, and (**D**) RDSVS-TCC cells after 24 h of treatment with increasing concentrations of Degrayn and PR-619. Actin serves as a loading control.

**Table 1 biomedicines-11-00759-t001:** Comparison of Degrasyn concentrations that inhibited 50% of the cell proliferation (IC_50_ values (μM)) for all tested cell lines after 24, 48, and 72 h of exposure, evaluated with MTT assay.

Time/Cell Line	SV-HUC-1	T24	K9TCC-PU-NK	RDSVS-TCC1
24 h	3.07 ^a^ ± 0.17	2.33 ^b^ ± 0.19	4.65 ^c^ ± 0.27	4.36 ^c^ ± 0.22
48 h	1.59 ^a^ ± 0.09	1.43 ^a^ ± 0.06	3.50 ^b^ ± 0.16	3.76 ^b^ ± 0.56
72 h	1.51 ^a^ ± 0.08	1.37 ^a^ ± 0.12	3.52 ^b^ ± 0.04	3.21 ^c^ ± 0.10

Note: The concentration range used was 0.6, 1.25, 2.5, 5, 10, and 20 μM. For each cell line, values without common letters (a, b, c) in the superscript differ statistically (*p* < 0.05). Results are presented as means ± standard deviations (SD) of three independent experiments (three wells each).

**Table 2 biomedicines-11-00759-t002:** Comparison of PR-619 concentrations that inhibited 50% of the cell proliferation (IC_50_ values (μM)) for all tested cell lines after 24, 48, and 72 h of exposure, evaluated with MTT assay.

Time/Cell Line	SV-HUC-1	T24	K9TCC-PU-NK	RDSVS-TCC1
24 h	7.28 ^a^ ± 0.12	6.78 ^a^ ± 0.78	13.19 ^b^ ± 0.58	13.41 ^b^ ± 0.56
48 h	4.23 ^a^ ± 0.11	3.70 ^a^ ± 0.37	7.19 ^b^ ± 0.26	8.56 ^b^ ± 1.58
72 h	3.85 ^a^ ± 0.06	3.23 ^a^ ± 0.31	6.15 ^b^ ± 0.60	7.31 ^b^ ± 0.61

Note: The concentration range used was 1.25, 2.5, 5, 10, and 20 μM. For each cell line, values without common letters (a, b) in the superscript differ statistically (*p* < 0.05). Results are presented as means ± standard deviations (SD) of three independent experiments (three wells each).

## Data Availability

The data supporting the results of this study can be obtained on request from the corresponding author.

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
