# Peer review of "Ubiquitin-Specific Proteases as Potential Therapeutic Targets in Bladder Cancer—In Vitro Evaluation of Degrasyn and PR-619 Activity Using Human and Canine Models"

_biomedicines, 2023, doi:10.3390/biomedicines11030759_

Round 1

Reviewer 1 Report

This research demonstrates that selective ubiquitin-specific protease inhibition seems to be more effective in reducing bladder cancer cell proliferation than non-selective ubiquitin protease inhibition.

Discussion may be revised to describe the differences between Degrasyn and PR-619 in terms of the specificity in cancer cells with additional references. 

Author Response

Dear Editor,

We are submitting a revised version of the article in response to the decision to perform major revisions to the biomedicines-2092404 manuscript. All issues raised by the academic editor and reviewers have been corrected or discussed. A detailed report on the amendments is presented below. If the reviewers request additional language corrections, we will immediately send our manuscript to MDPI English Editing Service.

#Reviewer 1

  1. Discussion may be revised to describe the differences between Degrasyn and PR-619 in terms of the specificity in cancer cells with additional references.

Our response: We highly appreciate the reviewer’s feedback and valuable comment. Additional sections were added to the discussion section (yellow highlights).

Reviewer 2 Report

The article by Łukasz Nowak et al. entitled Ubiquitin-specific proteases as a potential therapeutic target in bladder cancer – in vitro evaluation of Degrasyn and PR-619 activity using human and canine model” still raises the following issue.

1.    The abstract should be a single paragraph and should follow the style of structured abstracts, but without headings.

2.    qPCR data needs to be supported with western blotting to really see apoptosis and proliferation. Authors can show apoptosis by Annexin V staining to be consistent and confirm the preliminary qPCR data.

3.    Authors did not perform any in vivo experiments in the present study and did not get a consistent outcome in the T24 xenograft murine models

4.    Authors can perform cell cycle/apoptosis markers and cell proliferation markers (Ki67, PCNA, BrdU, etc.)  by using western blot and substantiate their outcome

5.    Significant grammar and typographical errors were found throughout the manuscript and should be corrected.

6.    In order to validate the consistency of antiproliferaive  properties of Degrasyn and PR-619, authors can perform at least two cell lines for each species.

7.    Authors did not perform a murine model to support their hypothesis. Still, this work is primitive and needs to explore the underlying mechanism

Author Response

Dear Editor,

We are submitting a revised version of the article in response to the decision to perform major revisions to the biomedicines-2092404 manuscript. All issues raised by the academic editor and reviewers have been corrected or discussed. A detailed report on the amendments is presented below. If the reviewers request additional language corrections, we will immediately send our manuscript to MDPI English Editing Service.

#Reviewer 2

  1. The abstract should be a single paragraph and should follow the style of structured abstracts, but without headings.

Our response: We highly appreciate the reviewer’s feedback and valuable comments. The abstract was corrected as suggested.

  1. qPCR data needs to be supported with western blotting to really see apoptosis and proliferation. Authors can show apoptosis by Annexin V staining to be consistent and confirm the preliminary qPCR data.

Our response: We think that additional qPCR data would be an unnecessary addition to the article with no significant impact on the presented data. Annexin V staining is the primary method of apoptosis evaluation in our laboratory because it is a widely used method in basic research.

  1. Authors did not perform any in vivo experiments in the present study and did not get a consistent outcome in the T24 xenograft murine models

Our response: In our study we intended to focus only on the in vitro evaluation of USP inhibitors. We added the results of additional experiments explaining the mechanisms of Degrasyn's action.

  1. Authors can perform cell cycle/apoptosis markers and cell proliferation markers (Ki67, PCNA, BrdU, etc.)  by using western blot and substantiate their outcome

Our response: The Ki-67 proliferation assay was additionally performed (Line: 151 – 163; Line 310 – 317)

  1. Significant grammar and typographical errors were found throughout the manuscript and should be corrected.

Our response: The manuscript was corrected in terms of grammar and typographical errors.

  1. In order to validate the consistency of antiproliferaive  properties of Degrasyn and PR-619, authors can perform at least two cell lines for each species.

Our response: As the T24 cell line is a well-known representative cell line of urothelial bladder cancer, we decided to use it in our research without any additional cell lines. Also, we used SV-HUC-1 cell line as control. In many in vitro studies on bladder cancer authors used this combination of cell lines. In our opinion, the introduction of additional cell line would not have significant impact on the current research.

  1. Authors did not perform a murine model to support their hypothesis.

Our response: We intended to focus only on the in vitro evaluation of USP inhibitors. We added the results of additional experiments explaining the mechanisms of Degrasyn's action.

Round 2

Reviewer 2 Report

The present form is accepted for publication